# Structure and conformational dynamics of *Clostridioides difficile* toxin A

Baohua Chen[1], Sujit Basak[2], Peng Chen[1], Changcheng Zhang[2], Kay Perry[3], Songhai Tian[4], Clinton Yu[1], Min Dong[4], Lan Huang[1], Mark E Bowen[2], Rongsheng Jin[1]

***Clostridioides difficile* toxin A and B (TcdA and TcdB) are two major virulence factors responsible for diseases associated with *C. difficile* infection (CDI). Here, we report the 3.18-Å resolution crystal structure of a TcdA fragment (residues L843–T2481), which advances our understanding of the complete structure of TcdA holotoxin. Our structural analysis, together with complementary single molecule FRET and limited proteolysis studies, reveal that TcdA adopts a dynamic structure and its CROPs domain can sample a spectrum of open and closed conformations in a pH-dependent manner. Furthermore, a small globular subdomain (SGS) and the CROPs protect the pore-forming region of TcdA in the closed state at neutral pH, which could contribute to modulating the pH-dependent pore formation of TcdA. A rationally designed TcdA mutation that trapped the CROPs in the closed conformation showed drastically reduced cytotoxicity. Taken together, these studies shed new lights into the conformational dynamics of TcdA and its roles in TcdA intoxication.**

## Introduction

*Clostridioides difficile* is a Gram-positive, spore-forming, anaerobic bacterium, which is a major cause of hospital-acquired diarrhea and pseudomembranous colitis and classified as an urgent antibiotic resistance threat by the Center for Disease Control and Prevention (CDC). Two homologous exotoxins, toxin A (TcdA) and B (TcdB), are the key virulence factors of *C. difficile*, leading to *C. difficile* infections (CDI) with variable clinical features including life-threatening pseudomembranous colitis (Kuehne et al, 2010; Leffler & Lamont, 2015; Aktories et al, 2017). It has been estimated that there are ~223,900 CDI cases, associated with at least 12,800 deaths, and $1 billion attributable healthcare costs in the United States in 2017 (CDC, 2019).

TcdA (~308 kD) and TcdB (~270 kD) belong to the large clostridial glucosylating toxin (LCGT) family, which also include *Paeniclostridium sordellii* toxins TcsL and TcsH, *Clostridium novyi* toxin TcnA,

and *Clostridium perfringens* toxin TpeL (Aktories et al, 2017; Orrell & Melnyk, 2021). Most of these toxins are composed of four structural modules: an N-terminal glucosyltransferase domain (GTD), followed by a cysteine protease domain (CPD), a delivery and receptor-binding domain (DRBD), and a large C-terminal combined repetitive oligopeptides domain (CROPs) (Aktories et al, 2017; Orrell & Melnyk, 2021) (Fig 1A). The cellular uptake of TcdA and TcdB are mediated by receptor-mediated endocytosis (Papatheodorou et al, 2010; Tao et al, 2016, 2019; Aktories et al, 2017; Chen et al, 2018, 2021). Triggered by acidification in the endosomes, the hydrophobic pore-forming region of the toxin undergoes membrane insertion and facilitates the translocation of the GTD and the CPD into the cytosol (Qa'Dan et al, 2000; Barth et al, 2001; Genisyuerek et al, 2011; Zhang et al, 2014). Activated by cytosolic inositol hexakisphosphate (InsP6), the CPD cleaves off the GTD and releases it into the cytosol (Egerer et al, 2007; Reineke et al, 2007). The GTD then glucosylates and inactivates the Rho and/or Ras families of small guanosine triphosphatases (GTPases) in host cells, resulting in depolymerization of the actin cytoskeleton, cell rounding, and ultimately cell death (Just et al, 1995a, 1995b; Chen et al, 2015; Liu et al, 2021).

Great efforts have been made to explore the structures of TcdA, and many fragment structures of TcdA have been reported, including crystal structures of its GTD, CPD, a CROPs-less fragment consisting of the GTD, CPD, and DRBD (residues 1–1,832, PDB code: 4R04), and several small fragments of the CROPs (Ho et al, 2005; Greco et al, 2006; Pruitt et al, 2009, 2012; Chumbler et al, 2016; Kroh et al, 2017). A recent cryoEM study reveals a near-complete TcdA structure including residues 2–2,383, but still missing part of its C-terminal CROPs (PDB code: 7POG) (Aminzadeh et al, 2022).

The CROPs domains are unique structural modules for toxins in the LCGT family, with TpeL being the only member that does not possess a CROPs (Aktories et al, 2017; Orrell & Melnyk, 2021). The CROPs of TcdA has seven units (CROPs I-VII) including 32 short repeats (SRs) and 7 long repeats (LRs) (Ho et al, 2005) (Figs 1B and S1A). The functional contribution of the CROPs to TcdA toxicity is evidenced by the observations that TcdA CROPs can bind to oligosaccharides on the host cell surface for cell entry (Krivan et al, 1986; Tucker & Wilkins, 1991; Teneberg et al, 1996), it protects the

[1]Department of Physiology and Biophysics, School of Medicine, University of California, Irvine, Irvine, CA, USA  [2]Department of Physiology and Biophysics, Stony Brook University, Stony Brook, NY, USA  [3]NE-CAT and Department of Chemistry and Chemical Biology, Cornell University, Argonne National Laboratory, Argonne, IL, USA  [4]Department of Urology, Boston Children's Hospital, Department of Microbiology and Department of Surgery, Harvard Medical School, Boston, MA, USA

Correspondence: r.jin@uci.edu; mark.bowen@stonybrook.edu

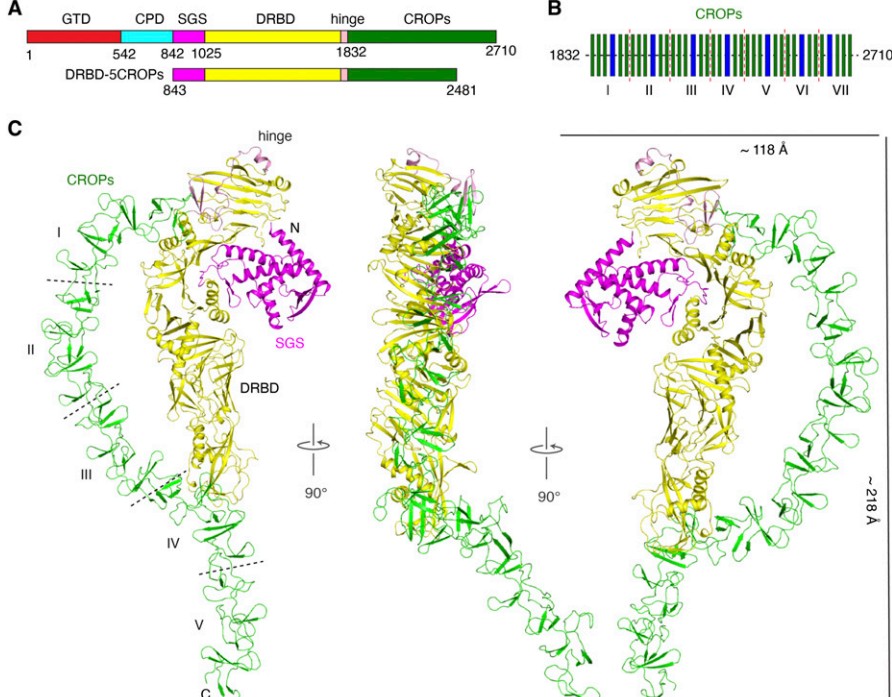

**Figure 1.   The overall architecture of DRBD-5CROPs of TcdA.**

**(A)** A schematic diagram of TcdA showing its domain organization and the fragment used for crystallization: GTD (red), CPD (cyan), SGS (magenta), DRBD (yellow), hinge (pink), and CROPs (green). **(B)** The CROPs of TcdA is composed of seven units (I to VII) including 32 short repeats (SRs, thin green bars) and 7 long repeats (LRs, thin blue bars). The dashed lines indicate the boundaries of CROPs I-VII. **(C)** Cartoon representations of DRBD-5CROPs with the components colored as in panel (A).

toxin from premature autoprocessing of the GTD and inactivation at neutral pH (Olling et al, 2014), and it is also the major target for several neutralizing antibodies including actoxumab that went to clinical trials (Hussack et al, 2011; Murase et al, 2014; Hernandez et al, 2017; Kroh et al, 2017). Another unique feature of the CROPs is that it can undertake large conformational changes in a pH-dependent manner that has been observed for both TcdA and TcdB (Pruitt et al, 2010; Chen et al, 2019). For example, a prior EM study showed that TcdA adopts a relatively compact closed conformation at neutral pH where its CROPs is attached to the DRBD, whereas at acidic pH the CROPs swings away from the DRBD and could adopt multiple orientations with respect to the rest of the toxin, leading to an open conformation (Pruitt et al, 2010). Therefore, it is crucial to understand how the CROPs of TcdA structurally interacts and functionally coordinates with the rest of the molecule in the context of the supertertiary structure of the holotoxin.

Here, we report a 3.18-Å resolution crystal structure of a TcdA fragment composed of DRBD and CROPs I–V (residues L843-T2481, referred to as DRBD-5CROPs), which serves as the foundation for us to piece together all the known fragment structures of TcdA into a complete structure of TcdA holotoxin. We also present complementary mutagenesis, limited proteolysis, and single molecule fluorescence resonance energy transfer (smFRET) studies to demonstrate that the CROPs of TcdA can dynamically sample open and closed conformations relative to the rest of the toxin in a pH-dependent manner, which could contribute to modulating the action of the pore-forming region in the DRBD, and that conformational dynamics is crucial for TcdA cytotoxicity. Furthermore, mapping the receptor-binding sites and neutralizing epitopes on the CROPs of TcdA in the context of holotoxin provides new insights

into the roles of the CROPs in TcdA intoxication and strategies for developing medical countermeasures against TcdA.

## Results

### The structure of the DRBD-5CROPs of TcdA

The holotoxin of TcdA has evaded structural studies due to its large molecular weight, multi-domain organization, high structural flexibility, and challenges in protein production and purification. When we initiated this project in 2018, the known crystal structure for the largest TcdA fragment includes residues 1–1,832 (referred to as TcdA[1832], PDB code: 4R04) without the CROPs (Chumbler et al, 2016), whereas the overall architecture of TcdA holotoxin was revealed by a negative stain EM study (Pruitt et al, 2010). In addition, several small fragment structures are reported for CROPs VI-VII (Greco et al, 2006; Murase et al, 2014; Kroh et al, 2017) or CROPs VII (Ho et al, 2005; Murase et al, 2014). Given the wealth of TcdA fragment structures and its modular architecture, we reasoned that the only missing piece in this puzzle needed to assemble the complete picture of TcdA holotoxin is the structure of a fragment covering the DRBD and the CROPs.

We thus examined multiple designs of TcdA (strain VPI10463) truncations, focusing on the DRBD and various lengths of the CROPs, for their recombinant expression profiles and biochemical behaviors. We found that the most suitable TcdA fragment for crystallization is composed of residues L843 to T2481 including the complete DRBD and CROPs I–V (referred to as DRBD-5CROPs) (Fig 1A). The best crystals of DRBD-5CROPs were obtained by

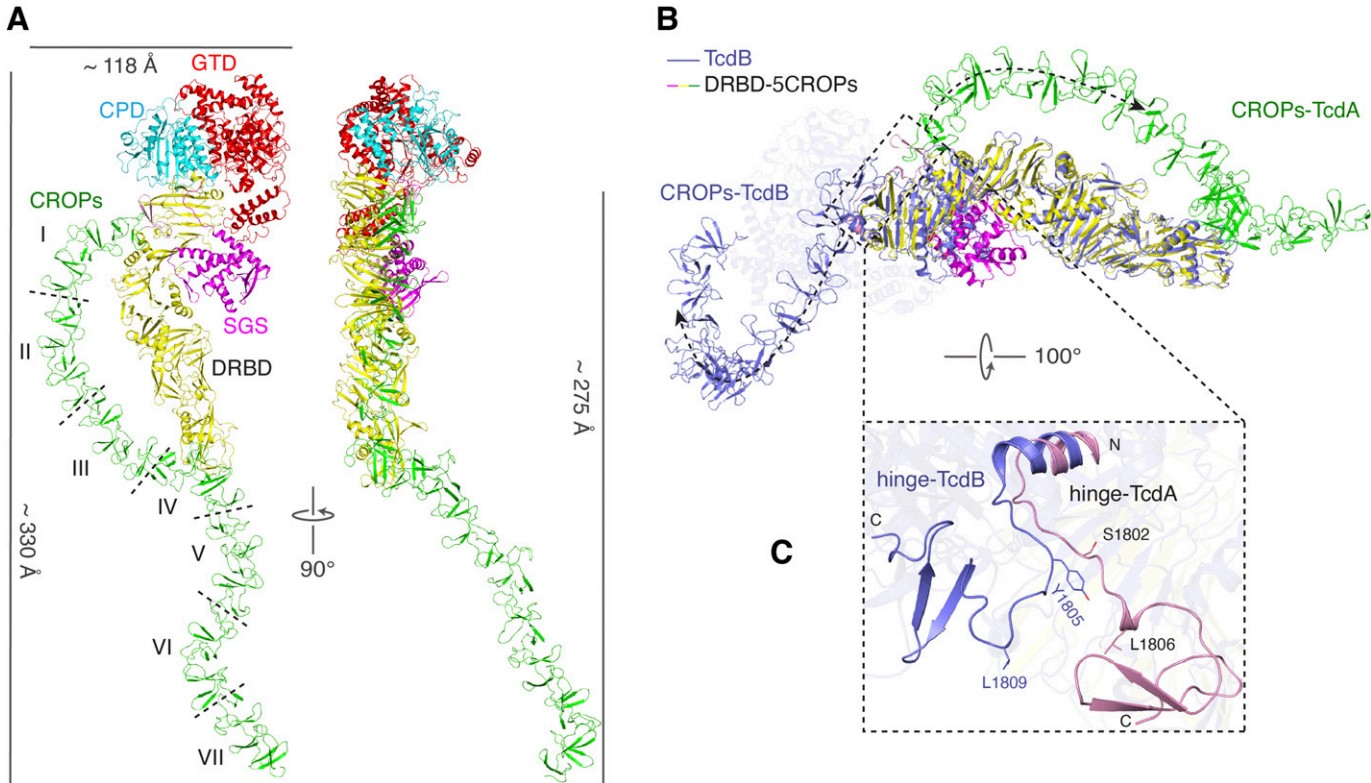

**Figure 2. A structure model of TcdA holotoxin.**
**(A)** Cartoon representations of the structure of TcdA holotoxin in two different views. The dashed lines indicate the boundaries of CROPs I-VII. **(B)** The superimposed structures of DRBD-5CROPs of TcdA (SGS: magenta; DRBD: yellow; CROPs: green) and TcdB holotoxin (blue) based on the DRBD. The dashed lines with arrows show different orientations of the CROPs of TcdA and TcdB. **(C)** A close-up view into the hinge region in the superimposed TcdA DRBD-5CROPs (pink) and TcdB holotoxin (blue).

streak-seeding in hanging-drops, and the best X-ray diffraction data we obtained was at 3.18-Å resolution (Table S1). The crystal structure of DRBD-5CROPs was solved by molecular replacement (see the Materials and Methods section), and a near complete structure including residues 855–2,481 was built except for a small flexible region including residues 1,661–1,667 that had no visible electron density (Fig 1C).

The structure of DRBD-5CROPs starts with an N-terminal small globular subdomain (SGS, residues 850–1,025) that is part of the DRBD but with a largely unknown function (Aktories et al, 2017) (Fig 1C). The SGS is also referred to as the globular subdomain (GSD) in the recent cryoEM study (Aminzadeh et al, 2022). Notably, a hydrophobic region buried in the SGS (residues 958–1,039) is considered to be the N-terminal portion of the pore-forming region (PFR, residues 958–1,130) on TcdA, and the rest of PFR then extends away from the SGS forming four α-helices (PFR-α1 to α4, residues 1,040–1,130) that stretch across the elongated DRBD. The PFR is involved in pH-dependent membrane insertion and delivery of the GTD and the CPD from endosomes to the cytosol (Qa'Dan et al, 2000; Barth et al, 2001; Genisyuerek et al, 2011; Zhang et al, 2014), whereas the hydrophobic residues in the PFR are protected by the DRBD at neutral pH and released for pore formation when induced by acidic endosomal pH (Chumbler et al, 2016; Aktories et al, 2017; Chen et al, 2019). The N-terminus of the CROPs is connected via a hinge region (residues 1,789–1,831) to the proximal tip of the DRBD where the SGS

is localized (Fig 1C). The CROPs I to IV then forms an arch-like structure pointing to the distal tip of the DRBD, where the first and second SRs and LR4 of CROPs IV interact with the DRBD. The rest of the CROPs IV-V kinks by ~35° and extends further away from the DRBD (Fig 1C).

## The overall architecture of TcdA holotoxin

Using the structure of DRBD-5CROPs (residues 843–2,481) as a basis, we were able to model the structures of residues 1–842 based on structure superposition with the structures of TcdA[1832] (PDB code: 4R04) (Chumbler et al, 2016), and build the complete CROPs based on superposition with the structure of a CROPs VI-VII fragment (residues 2,456–2,710) (PDB code: 2G7C) (Greco et al, 2006), resulting a complete structural model of TcdA holotoxin (Fig 2A). We found that the structures of the GTD, CPD, DRBD, and the CROPs I-III in our TcdA holotoxin model fit well to a low resolution negative stain EM map of TcdA, but the C-terminal CROPs IV-VII kinked by ~35° compared with the EM map (Pruitt et al, 2010), indicating that the EM density in this region is likely an artifact due to flattening of negative stained specimens (Fig S1B and C). Our TcdA holotoxin structure model is also consistent with a 2.8-Å resolution cryoEM structure of TcdA (residues 2–2,383), which was reported during the preparation of this manuscript (Fig S1D) (Aminzadeh et al, 2022).

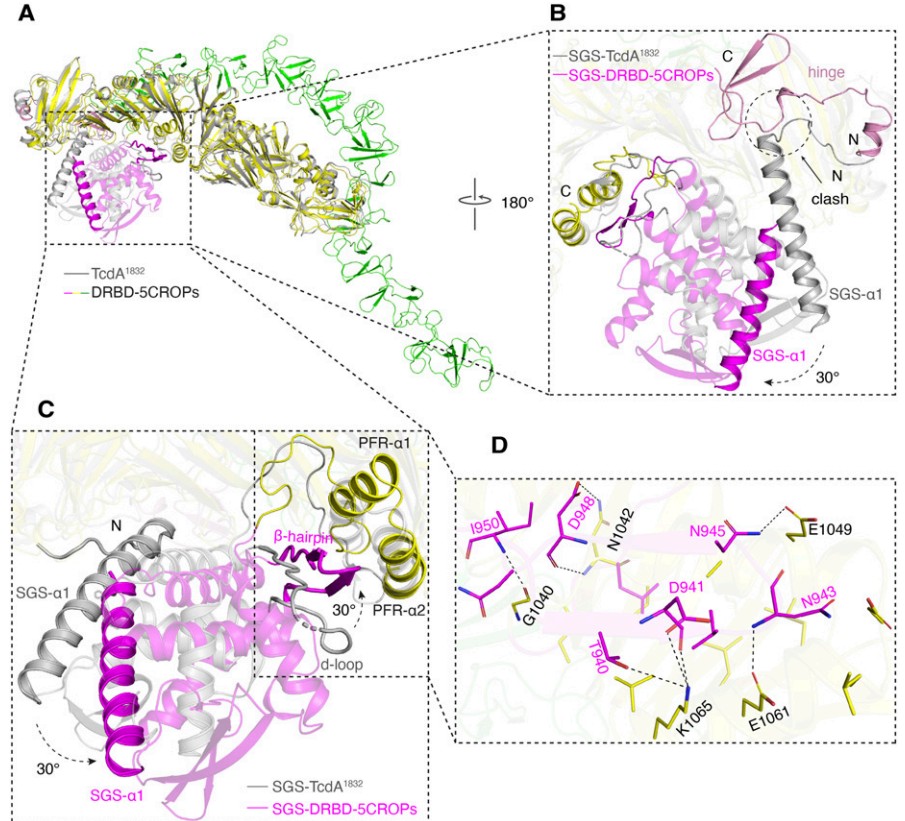

**Figure 3. The conformation of the small globular subdomain (SGS) is correlated to that of the CROPs.** **(A)** The superimposed structures of DRBD-5CROPs (SGS: magenta; DRBD: yellow; CROPs: green) and TcdA[1832] (gray). The GTD and CPD of TcdA[1832] are not shown for clarity. **(B)** A close-up view of the SGS around the hinge region. The SGS observed in TcdA[1832] would clash with the hinge in the context of DRBD-5CROPs. SGS-α1 consists of residues 852–878. **(C)** The SGS shows a ~30° rotation in the structures of DRBD-5CROPs when compared with TcdA[1832]. Residues 937–953 is a partly disordered loop in TcdA[1832] (d-loop), but forms a well-defined β-hairpin in DRBD-5CROPs that interacts with PFR-α1 and α2. **(D)** A close-up view into the interface between the β-hairpin and PFR-α1 and α2 in DRBD-5CROPs with interacting amino acids shown in stick models.

It is worth noting that the crystal structure of DRBD-5CROPs and the cryoEM structure of TcdA were both obtained at near neutral pH. In both cases, the CROPs emerges from the junction of the GTD, CPD, and DRBD, lies parallel to the DRBD, touches the distal tip of the DRBD, and then continues extending further away, forming a closed conformation (Fig 2A). Nevertheless, earlier negative stain EM studies showed that this conformation of TcdA was only observed at neutral pH, whereas the CROPs detached and moved away from the DRBD at acidic pH to adopt an open conformation (Pruitt et al, 2010). Whereas the open conformation of TcdA has not been structurally defined, it is believed that it would be analogous to the open conformation of TcdB holotoxin, which represents an acidic-pH specific conformation where the CROPs of TcdB adopts a hook-like structure and swings to the opposite direction of the DRBD (Fig 2B) (Chen et al, 2019).

We therefore compared the closed conformation of TcdA with the open conformation of TcdB to explore how the CROPs can adopt such a drastic conformational change. We found that the rotation of the CROPs is largely mediated by a flexible loop (residues 1,797–1,811 in TcdA and 1,800–1,814 in TcdB, referred to as the hinge loop) in the hinge, whereas an α-helix to the N terminus of the hinge loop attaches to the DRBD and a downstream β-hairpin links to the CROPs (Figs 2C and S2A). More specifically, the hinge loop adopts a relatively linear structure in the closed conformation of TcdA, but forms a U-shape structure in TcdB to yield the open conformation. The two ~90° turns in the hinge loop of TcdB happen around residues Y1805 and L1809,

which are structurally equivalent to residues S1802 and L1806 on TcdA (Fig 2C).

## The SGS in the DRBD is structurally coordinated with the CROPs

The SGS is a well folded structural unit located at the N-terminus of the DRBD, which is composed of five α-helices connected by six-stranded β-sheets and several loops, but its function remains unknown (Aktories et al, 2017). We observed a large ~30° rotation of the SGS relative to the rest of the DRBD when comparing the structures of DRBD-5CROPs and the CROPs-less TcdA[1832] (Chumbler et al, 2016) (Fig 3A–C), although the overall folds of the DRBD in these two structures are similar (root-mean-square deviation, RMSD, of ~1.9 Å over 722 superimposed Cα atoms). The conformation of the SGS is similar in DRBD-5CROPs and the recent cryoEM structure of TcdA holotoxin, but the SGS in the open TcdB holotoxin adopts a conformation similar to that of TcdA[1832] (Fig S3A).

In DRBD-5CROPs of TcdA, the SGS moves closer to PFR-α1 and α2. This movement is likely needed to accommodate the closing of the CROPs, as the SGS will otherwise clash with the "closed" hinge in DRBD-5CROPs (Figs 3A and C and S3A and B). Noticeably, residue D1826 in the "closed" hinge forms a hydrogen bond with H954 in the SGS, which further stabilize the matching conformation of SGS (Fig 4). In comparison, the conformation of the SGS in the open TcdB fits well with the "open" hinge that allows the CROPs to take the open conformation (Fig S3A). We suspect that, in the absence of the CROPs, the SGS in TcdA[1832] likely takes the open conformation,

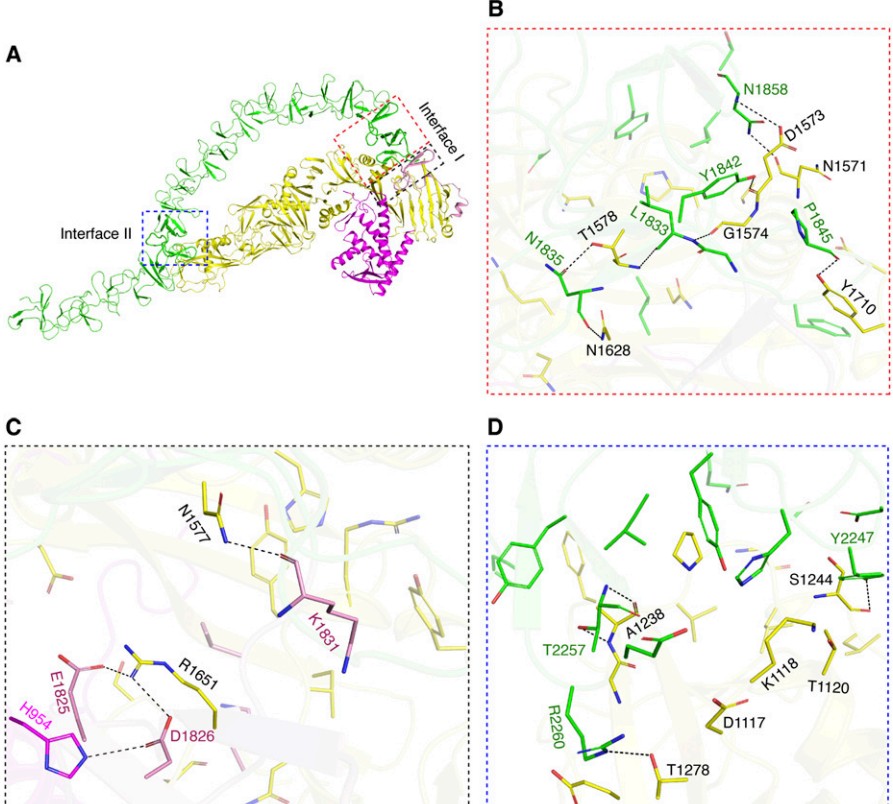

**Figure 4. Interactions between the DRBD and the CROPs in the closed conformation of TcdA.**
**(A)** A cartoon representation of DRBD-5CROPs showing the interface I and II. **(B, C, D)** Close-up views into the interactions at interface I between the CROPs and DRBD (B), among the hinge, SGS, and DRBD (C), and at interface II between the CROPs and DRBD (D).

which is similar to that of TcdB in the open conformation. These findings suggest that the SGS rotation is likely coordinated with the movement of the CROPs in TcdA and TcdB.

We found that a well-defined β-hairpin in the SGS of DRBD-5CROPs (residues 937–953) directly interacts with PFR-α1 and α2 (Figs 3C and D, S2B, and Table S2). This β-hairpin was also revealed in the recent cryoEM studies and named the "guard-loop" (Aminzadeh et al, 2022). It is well accepted that the PFR is shielded by the DRBD at neutral pH, which undergoes large conformational changes induced by acidic endosomal pH to facilitate the delivery of the GTD and the CPD across the membrane. As revealed by a structure of TcdB that was obtained under acidic pH and represents a pore-forming intermediate state, PFR-α1 and α2 partially unfold and detach from the DRBD at acidic pH (Fig S3C) (Chen et al, 2019). We envision that the β-hairpin in the SGS in the closed form of TcdA can help to stabilize PFR-α1 and α2 in the DRBD-shielded conformation at neutral pH. Notably, this β-hairpin turns into a partly disordered loop in TcdA[1832] (referred to as d-loop), which shifts away from the PFR and loses contact with PFR-α1 or α2 (Figs 3C and S3B). Moreover, the region equivalent to the TcdA β-hairpin in the acidic open conformation of TcdB (residues 937–953) is also disordered in a way similar to that observed in TcdA[1832] (Fig S3B). These findings suggest that the conformations of the SGS and this β-hairpin are structurally coordinated with the movement of the CROPs in a pH-dependent manner. The β-hairpin in the SGS in the context of the closed conformation of the CROPs directly interacts with PFR-α1 and α2 helices and helps to stabilize the PFR in the

inactive and shielded state at neutral pH, whereas the SGS will move away and the β-hairpin will unfold and release the PFR when the CROPs takes the acidic pH-specific open conformation.

## The interplay between the DRBD and the CROPs of TcdA

In the closed conformation of TcdA, the CROPs interact with the DRBD at two distinct regions (Fig 4A). The N terminus of the CROPs is linked to the DRBD via the hinge region, and the hinge and CROPs establish extensive interactions with the DRBD in this area (referred to as interface I). More specifically, residues E1825, D1826, and K1831 in the hinge form hydrogen bonds with residue H954, N1577, and R1651 in the DRBD; residues E1825 and D1826 in the hinge form salt bridges with R1651 in the DRBD; residues L1833, N1835, Y1842, and P1845 in the first SR of CROPs I form hydrogen bonds with D1573, G1574, T1578, N1628, and Y1710 in the DRBD; and residue N1858 in the second SR of CROPs I form hydrogen bonds with N1571 and D1753 in the DRBD (Fig 4B and C and Table S3). These interactions stabilize the CROPs in the closed conformation at the proximal tip of the DRBD.

The second interface is established between the distal tip of the DRBD and the CROPs IV (referred to as interface II) (Fig 4A). The interactions at interface II are relatively weak, which are mainly mediated by van der Waals interactions supplemented with four pairs of hydrogen bonds between residues Y2247, T2257, and R2260 in the CROPs and A1238, S1244, and T1278 in the DRBD (Fig 4D and Table S4). A similar interface was reported in the recent cryoEM

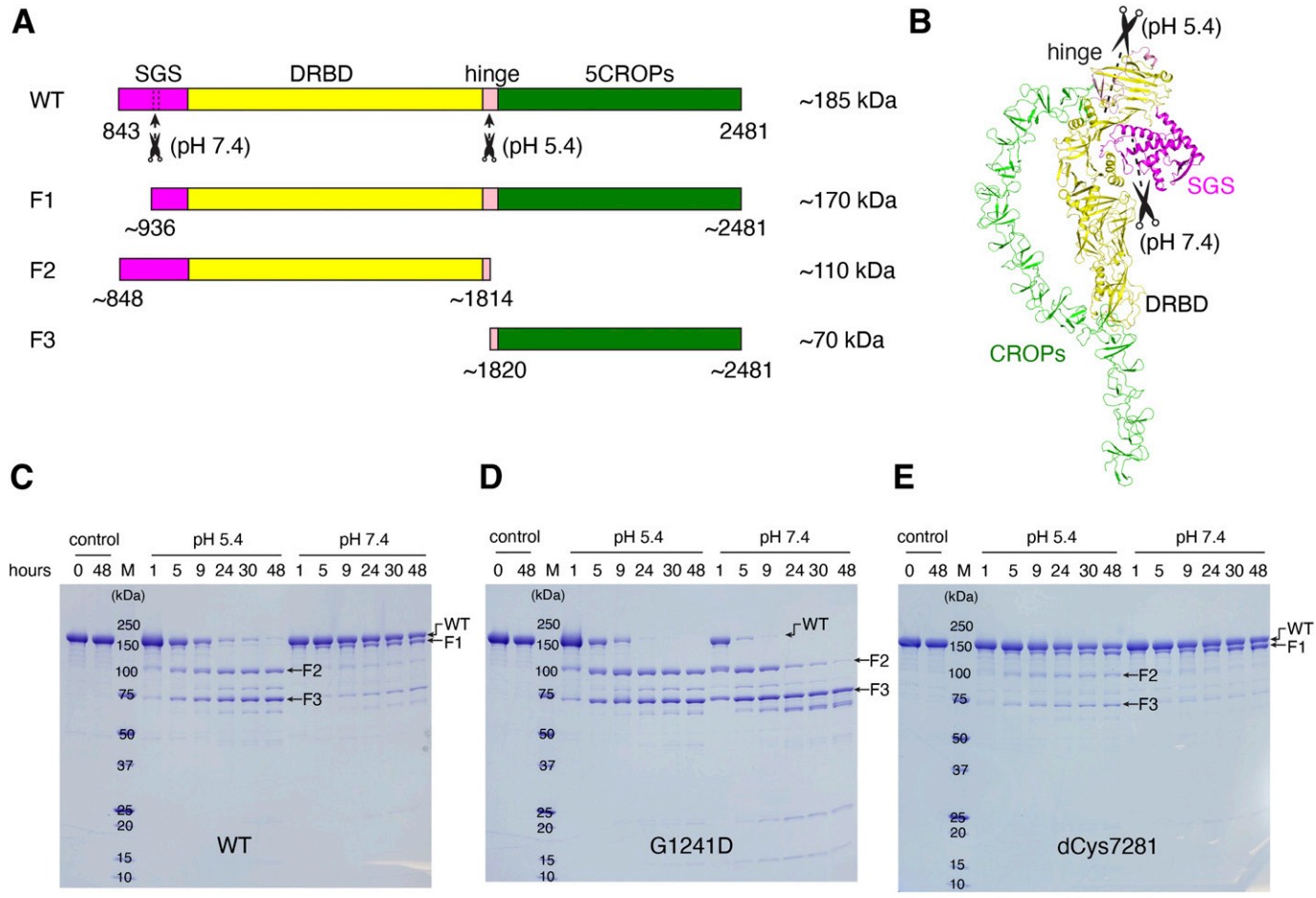

**Figure 5. Examining the conformational flexibility of DRBD-5CROPs using limited trypsin proteolysis.**
**(A)** A schematic diagram of DRBD-5CROPs and three major cleavage products (F1, F2, and F3). **(B)** Two estimated trypsin cleavage sites on DRBD-5CROPs. **(C, D, E)** Limited trypsin digestion was performed on the wild-type DRBD-5CROPs (WT), DRBD-5CROPs$^{G1241D}$ (G1241D), and DRBD-5CROPs$^{dCys7281}$ (dCys7281) at pH 5.4 and pH 7.4. Shown are representative SDS–PAGE gels with Coomassie blue staining from three independent experiments.

studies except for some subtle differences in side chain orientation, which may be due to the different conditions and methods used for structure determination (Aminzadeh et al, 2022). Interestingly, we found that three residues in PFR-α4 (D1117, K1118, and T1120) interact with H2234, T2247, and E2259 in the CROPs IV via van del Waals interactions (Fig 4D and Table S4), suggesting that the CROPs can help stabilize the pore-forming region at interface II in the closed conformation. Nevertheless, the interface II is relatively small that buries a molecular surface area of ~623.8 Å$^2$ per molecule, making it amenable for the CROPs to reversibly associate or dissociate with the DRBD.

### pH-dependent dynamics of TcdA

All the LCGT family members possess the CROPs except for TpeL (Schorch et al, 2014), and the CROPs can undergo pH-dependent movement (Pruitt et al, 2010; Chen et al, 2019). To further characterize the structural flexibility of the CROPs, we resorted to limited trypsin proteolysis, which can detect protein structural rearrangements based on the change of cleavage patterns

(Schopper et al, 2017). We focused our studies on DRBD-5CROPs because this fragment has high expression yield, is well folded and more accessible for biochemical and mutagenesis studies than TcdA holotoxin.

Limited trypsin digestion was performed on DRBD-5CROPs at pH 7.4 and pH 5.4 over a period of 48 h (Fig 5A–C). We found that DRBD-5CROPs was relatively resistant to trypsin at pH 7.4, and there was one major cleaved fragment that is slightly smaller than the full length DRBD-5CROPs (referred to as F1). However, DRBD-5CROPs was almost completely degraded by 9–24 h at pH 5.4, yielding two major cleavage fragments that are ~110 and ~70 kD, respectively (referred to as F2 and F3) (Fig 5C). These results demonstrate that DRBD-5CROPs adopts distinct conformations at these two different conditions, which expose different sites that are accessible for trypsin cleavage. We then performed mass spectrometry studies to identify these trypsin cleavage sites, which revealed that F1 is likely composed of residues ~936–2,481, whereas F2 and F3 are composed of residues ~848–1,814 and ~1820–2,481, respectively (Figs 5A and B and S4). These results suggest that there are two major sites on DRBD-5CROPs that are sensitive to trypsin digestion. One is located

in the SGS that is more exposed at neutral pH, but partly masked at acidic pH, whereas the other one is in the hinge, which is more accessible at acidic pH, but not at neutral pH.

Based on these findings, we hypothesized that the closed conformation of the CROPs at neutral pH may mask the trypsin site in the hinge and expose the site in the SGS, whereas the open conformation of the CROPs at acidic pH will expose the trypsin site in the hinge and mask the site in the SGS. To validate this hypothesis, we designed two different types of DRBD-5CROPs mutants based on its structure to disrupt or strengthen interactions between the DRBD and the CROPs at interface II. Mutant DRBD-5CROPs[G1241D] carries G1241D mutant in the DRBD, which will disrupt the interactions at interface II. Mutant DRBD-5CROPs[dCys7281] carries two cysteine at F1272C in the DRBD and P2281C in the CROPs, which will form a disulfide bond to cross link the DRBD and the CROPs in the closed state. The expression level and the purity of these two mutants of DRBD-5CROPs were similar to the WT protein, and they were mono-dispersed in solution as judged by the size-exclusion chromatography, suggesting their proper protein folding. Furthermore, we determined the crystal structure of DRBD-5CROPs[dCys7281] at 4.01-Å resolution, which confirmed the formation of a disulfide bond between F1272C and P2281C (Fig S2C and D), which locks the DRBD and the CROPs at interface II.

We then performed limited trypsin digestion on these two mutants and found that these two proteins lost their sensitivity to environmental pH changes. More specifically, DRBD-5CROPs[G1241D] turned more sensitive to trypsin even at pH 7.4, and its digestion patterns at pH 5.4 and 7.4 were both similar to that of the WT protein at pH 5.4 (Fig 5D). In contrast, DRBD-5CROPs[dCys7281] was more resistant to trypsin at pH 5.4, whereas its digestion patterns at pH 5.4 and pH 7.4 were both similar to that of the WT protein at pH 7.4 (Fig 5E). Taken together, these results suggest that the pH-dependent dynamics of TcdA is partly mediated by the interactions of the DRBD and the CROPs at interface II, and that adopting the closed conformation could be advantageous for the toxin to better resist proteases. Triggered by the endosomal acidic pH, the CROPs will detach from the DRBD, which may prime the toxin for the subsequent actions.

## Single-molecule FRET analyses of TcdA

To further probe the conformational dynamics of TcdA, we performed smFRET studies (Hellenkamp et al, 2018; Chen et al, 2019). We used the same DRBD-5CROPs fragment from the crystallographic study because TcdA holotoxin contains seven cysteine residues, which complicates site-specific dye conjugation. Guided by the crystal structure of DRBD-5CROPs, we created two FRET variants aiming to monitor the movement of the CROPs relative to the DRBD with one dye conjugated to the CROPs and the other on the DRBD. More specifically, we first created a cysteine-free DRBD-5CROPs with all endogenous cysteine residues mutated into serine. We then reintroduced one native cysteine with one cysteine mutation to create a FRET pair (Fig 6A). The SC variant (S1255C-C2236) spans the interface II that maintains the CROPs in the closed state, and the two labeling sites were selected not to disrupt the contacting interface. The EC variant (E1446C-C2023) probes the movement of the middle part of the CROPs relative to the DRBD.

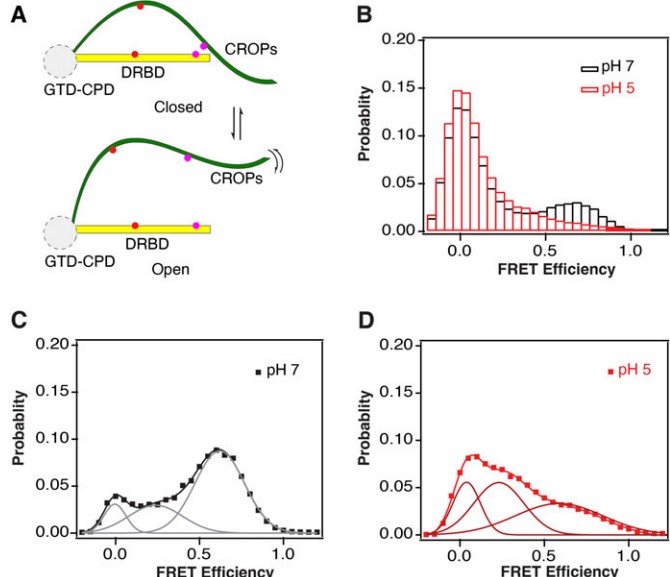

**Figure 6. smFRET analysis of the pH-dependent dynamics of DRBD-5CROPs.**
**(A)** A cartoon showing the representative closed and open conformations of TcdA and the relative locations of the labeling sites. The engineered Cys residues for dye conjugation in the SC (S1255C/C2236) and EC (E1446C/C2023) variants are shown as magenta and red dots, respectively. **(B)** Population histograms of per-frame FRET efficiency for all SC variant molecules under different pH: pH 7.0 (black) and pH 5.0 (red). **(C, D)** Population histograms of FRET efficiency for FRET-containing SC variant molecules. **(C)** pH 7.0 (black) with three Gaussian fit with low, intermediate and high FRET state (gray), and (D) pH 5.0 (red); fit (maroon).

Based on the structure of DRBD-5CROPs at neutral pH, the distances between the Cα atoms of the residues used for labeling were 3–4 nm. This should yield high FRET, and movement of the CROPs would affect energy transfer between the two dyes.

Both the SC and the EC variants of DRBD-5CROPs were expressed and purified similarly to the WT protein, and they were randomly labeled with a 1:1 mixture of Alexa Fluor 555 and Alexa Fluor 647 to greater than 90% efficiency. Proteins were chemically biotinylated targeting the N terminus, and attached to a passivated microscope slide via streptavidin. Such direct attachment of proteins to a surface is not ideal for studying protein dynamics but provides a cursory assessment of the conformational distribution. Proteins were imaged using prism TIRF at 10 Hz with alternating laser excitation to identify molecules containing one donor and one acceptor fluorophore based on single step photobleaching.

The accumulated histograms of FRET efficiency for the entire population reveal the time-averaged distribution of states present in the surface-attached protein. The FRET histogram for SC at pH 7 showed a broad distribution with a predominant peak at zero FRET along with a broad peak at higher FRET (Fig 6B). Examination of time traces for individual molecules revealed that most molecules remained in the zero FRET state. Only ~30% of molecules sampled higher FRET efficiency with many showing visible dynamic transitions. The EC variant showed overwhelmingly zero FRET at pH 7 with only 13% of molecules sampling higher FRET (Fig S5A). It is possible that molecules, which never sample FRET at pH 7, may be adversely affected by the surface-attachment strategy because EM and X-ray

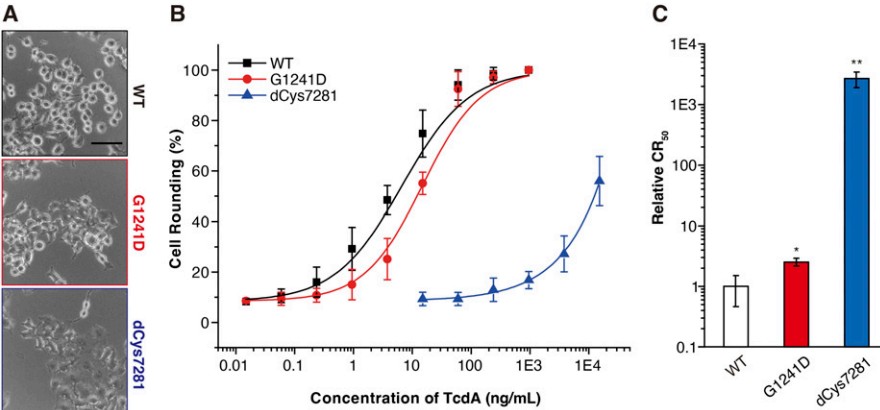

**Figure 7. The conformational dynamics of TcdA CROPs is crucial for its cytotoxicity.**
**(A)** Representative images showing the cell rounding effect of TcdA variants on HeLa cells after cells were treated with 4 ng/ml of TcdA$^{WT}$ (WT), TcdA$^{G1241D}$ (G1241D), and TcdA$^{dCys7281}$ (dCys7281) for 24 h. Scale bar, 20 $\mu$m. Representative images were from one of three independent experiments. **(B)** The percentages of round shaped HeLa cells after being treated with TcdA and variants for 24 h were plotted over toxin concentrations. Error bars indicate mean ± s.d., N = 3. **(C)** The toxin concentrations that induce half of cells to be round (CR$_{50}$) were quantified. The relative CR$_{50}$ values in treatment with TcdA variants were normalized to the WT TcdA and plotted as a bar chart. Error bars indicate mean ± s.d., N = 3. *$P < 0.05$; **$P < 0.01$ ($t$ test).

crystallographic studies have only observed the closed conformation at neutral pH.

To further investigate the high FRET states at pH 7, we selected out those molecules sampling higher FRET during the 50 s observation window. The time-averaged distributions for FRET-sampling molecules from both SC and EC variants were well fit to a three-state model (Figs 6C and S5B). The SC variant still showed a predominant peak at 0.62 (60% occupancy) with 20% occupancy of zero FRET along with a broad peak centered at 0.24 (20%) (Table S5). In contrast, the EC variant showed a predominant peak at zero FRET (60% occupancy) with a similar intermediate FRET peak to SC at 0.17 (30%), but a reduced high FRET peak (10%) (Table S5), which was extremely broad.

When the surface-attached SC variant was exposed to pH 5, we observed 60% decrease in the number of molecules sampling high FRET (13%) (Fig 6B). The population histogram for SC molecules sampling FRET at pH 5 revealed a predominant peak at zero FRET with a concomitant increase in intermediate FRET (Fig 6D and Table S5). The FRET-sampling of the EC variant also showed an increase in zero FRET with 20% decreases in intermediate and high FRET (Fig S5C and Table S5). Thus, both variants converged to a similar distribution at pH 5.

To extract rate constants from the dynamic FRET transitions, we used the ebFRET software package (van de Meent et al, 2014). We selected only those molecules sampling higher FRET during single molecule time traces for this analysis, which were the minority of molecules observed. For the SC variant at pH 7, the high FRET state was the longest lived with a dwell time of 2.9 s. The intermediate FRET state was the next longest lived at 0.8 s, whereas the zero FRET state dwell time was only 0.5 s (Fig S6A and Table S6). At pH 5, the high FRET state was destabilized with an 80% decrease in the mean dwell time in high FRET to 0.5 s. The intermediate state also decreased by half, whereas the low FRET dwell time increased by over 80% to 0.9 s (Fig S6B and Table S6). The EC variant showed similar trends with a similar mean dwell time in high FRET of 2.6 s. Most of these molecules were in stable high FRET with the dwell time limited by photobleaching. At pH 5, the EC variant dwell time in high FRET decreased by half, whereas the dwell time at zero FRET doubled (Fig S6C and D and Table S6). For both variants, molecules showing high FRET at pH 5 tended to be static, which was limiting the mean dwell times at high FRET.

Thus, the smFRET measurements suggest that when surface-attached, DRBD-5CROPs exists primarily in a low FRET state but dynamically samples a high FRET closed state, which tends to be static at neutral pH. We envision that one of the dominant closed conformations of TcdA at neutral pH is captured by the crystal structure reported here and a recent cryo-EM structure (Aminzadeh et al, 2022). The high FRET state is largely eliminated at pH 5, suggesting that TcdA may dynamically sample an ensemble of open conformations at acidic pH when the CROPs dissociates from the DRBD, which need to be further characterized in future studies.

### The dynamic conformation of TcdA CROPs is crucial for its cytotoxicity

Given our extensive structural and biochemical data demonstrating that the CROPs of TcdA display pH-dependent dynamics, we sought to determine how this unique feature contributes to TcdA cytotoxicity. To this end, we designed two mutants of TcdA holotoxin, TcdA$^{G1241D}$ carries the G1241D mutation and TcdA$^{dCys7281}$ carries F1272C and P2281C mutations, whose interactions between the DRBD and the CROPs at interface II are disrupted or strengthened, respectively. We then examined the cytopathic effects of these TcdA mutants using a cell-rounding assay, as inactivation of Rho GTPases by TcdA damages the actin cytoskeleton and results in the characteristic cell rounding phenotype (Hall, 1998) (Fig 7A). We found that the toxicity of TcdA$^{G1241D}$ on HeLa cells showed a slight ~2.5-fold reduction compared with TcdA$^{WT}$, whereas the toxicity of TcdA$^{dCys7281}$ drastically decreased by nearly 2,600-fold (Fig 7B and C). These data suggest that interactions between the DRBD and the CROPs at interface II is not essential for TcdA action, which may merely provide structural supports for TcdA in neutral environmental pH. In contrast, timely dissociation of the CROPs from the DRBD at acidic pH is crucial for TcdA cytotoxicity.

## Discussion

Here we determined the crystal structure of DRBD-5CROPs of TcdA, which we used as the foundation to build a model for the complete TcdA holotoxin. Whereas the structure of the toxin core composed

of the GTD, CPD, and DRBD is similar between TcdA and TcdB (Chen et al, 2019), the CROPs of TcdA adopts a closed conformation in the neutral crystallization condition that is in sharp contrast to the open conformation of the CROPs observed in a TcdB holotoxin structure that was obtained at acidic pH (Chen et al, 2019). More specifically, the hinge and the N-terminus of the CROPs form extensive interactions with the DRBD at the junction of the GTD, CPD, and DRBD (interface I). The elongated CROPs then extend like an arch toward the distal tip of the DRBD, where the CROPs IV attaches to the DRBD via a small interface (interface II), whereas the rest of the CROPs continues extending further away from the DRBD.

Structural analysis showed that the SGS in the DRBD undergoes a coordinated conformational change together with the CROPs. With the CROPs in the closed conformation, the SGS is located closer to the pore-forming region where the β-hairpin in the SGS interacts with PFR α1 and α2 helices. Our structure model suggests that, in the open conformation, the SGS will rotate ~30° away and lose contacts with the PFR. Furthermore, the CROPs in the closed conformation directly interacts with the C-terminal part of PFR through interface II. Therefore, TcdA in the closed conformation will likely stabilize the PFR in the buried, inactive conformation by simultaneously interacting with two distant parts of the PFR via the β-hairpin in the SGS and the CROPs IV.

Based on limited proteolysis and smFRET, we found that the CROPs adopts a spectrum of dynamic conformations, which is prone to form a closed conformation at neutral pH and undefined open conformations at acidic pH. The overall structure of TcdA is likely more stable in the closed conformation, which helps TcdA to better protect the PFR and resist environmental proteases at neutral pH. Prior studies also showed that TcdA is more resistant to autoprocessing of the GTD and inactivation at neutral pH (Olling et al, 2014). In contrast, an open conformation triggered by endosomal pH is necessary for the transition of TcdA to the next stage of cell invasion, as locking the CROPs in the closed conformation by a rationally designed disulfide bond greatly reduced its toxicity.

Whereas the CROPs of both TcdA and TcdB can undertake pH-dependent movement, only the closed conformation of TcdA and the open conformation of TcdB have been defined. A closed conformation of TcdB was captured by cross-linking mass spectrometry (XL-MS) in our prior studies, which showed that the C-terminal tip of the CROPs around residues K2234 in CROPs III and K2249 in CROPs IV can move within ~30 Å of the DRBD and be cross-linked to residues K1117, K1120, and K1126 on the DRBD (Fig S7) (Chen et al, 2019). When we built a model for the closed conformation of TcdB by superimposing the CROPs I-III of TcdB onto the TcdA holotoxin, we found that the hypothetic CROPs-DRBD interface in TcdB is likely overlapping with the interface II of TcdA (Fig S7A and B). This model of TcdB in a TcdA-like closed conformation is consistent with the results of XL-MS study and a similar modeling study based on a cryoEM structure of TcdA (Chen et al, 2019; Aminzadeh et al, 2022). Nevertheless, the CROPs IV of TcdB would clash with the distal tip of the DRBD in this model, whereas the CROPs IV of TcdA takes a ~35° rotation to establish the interface II (Fig S7B). How the CROPs III-IV of TcdB will reorient in order to properly interact with the DRBD in the closed conformation is well worth further studies.

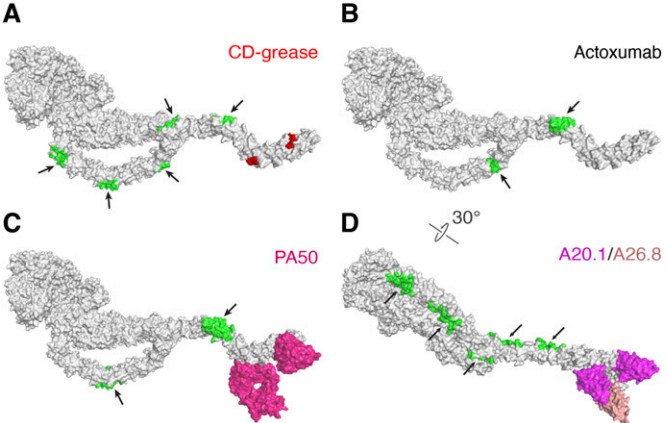

**Figure 8. Mapping the binding sites for the glycan receptor and selected neutralizing antibodies on TcdA holotoxin.**
**(A)** TcdA holotoxin is shown as a gray surface model. Two molecules of CD-grease (red sphere) are mapped to CROPs VI-VII in a crystal structure, whereas the putative receptor-binding sites on CROPs I–V are colored in green and indicated by black arrows. **(B)** Two putative epitopes (green) for actoxumab were identified in CROPs III and V. **(C)** Two epitopes were identified for PA50 (surface model in hotpink) on CROPs VI and VII as revealed by a crystal structure, and two additional putative epitopes (green) were proposed to locate in CROPs II and V. **(D)** Two epitopes were identified for VHH A20.1 (surface model in magenta) on CROPs VI and VII, respectively, and five additional putative epitopes (green) may locate on CROPs I to V. One epitope for VHH A26.8 (surface model in salmon) was identified on CROPs VII.

Prior studies suggest that TcdA CROPs can bind to oligosaccharides on the host cell surface for cell entry, although the native carbohydrate receptors on human intestinal epithelium has not been identified (Krivan et al, 1986; Tucker & Wilkins, 1991; Teneberg et al, 1996). With the complete structure of TcdA and its CROPs reported here, we are able to map these epitopes in the context of the holotoxin. A prior study showed that a derivative of a potential carbohydrate receptor for TcdA, α-Gal-(1,3)-β-Gal-(1,4)-β-GlcNA-cO(CH$_2$)$_8$CO$_2$CH$_3$ (CD-grease), can bind to two sites located in CROPs VI and VII, which could potentially bind to CROPs I–V too because the glycan-binding epitope is conserved in all seven CROPs units (Greco et al, 2006) (Fig 8A). These epitopes are all mapped on the toxin surface, and the flexibility of the CROPs may facilitate multivalent binding of TcdA to glycan receptors on cell surface.

Furthermore, the CROPs of TcdA is also the major target for several neutralizing antibodies including mAb actoxumab that was tested in clinical trials, which is in contrast to TcdB, whereas most of the known TcdB-neutralizing antibodies bind outside the CROPs (Chen & Jin, 2021), further emphasizing the functional roles of the TcdA CROPs. Because of the repetitive sequences in the CROPs, two epitopes were identified for actoxumab in CROPs III and V (Hernandez et al, 2017), four epitopes were identified for another mAb PA50 in CROPs II, V, VI, and VII with the latter two confirmed by a crystal structure (Kroh et al, 2017), seven epitopes were identified for VHH A20.1 across all 7 CROPs units with the two epitopes in CROPs VI and VII confirmed by crystal structures (Hussack et al, 2011; Murase et al, 2014), and one epitope was identified in CROPs VII for VHH A26.8 (Hussack et al, 2011; Murase et al, 2014) (Fig 8B–D). In general, antibody binding to the CROPs will generate physical barriers, preventing it from recognizing cell surface receptors,

which is consistent with observations that actoxumab and PA50 interfered with cell surface binding of TcdA (Hernandez et al, 2017; Kroh et al, 2017). Based on the structure of TcdA holotoxin, our structural modeling suggests that antibodies binding near CROPS IV may interfere with DRBD-CROPs interactions within interface II, whereas antibodies bound in CROPS I-IV may also restrict the movement of the CROPs due to potential physical clashes of the bound antibodies with the DRBD. This finding suggests new avenues to neutralize TcdA by restricting its conformational dynamics. The structure of TcdA holotoxin should provide a framework for future studies to further characterize the structural and functional roles of the CROPs in TcdA intoxication, to better understand the neutralizing mechanisms for therapeutic antibodies, and to develop new strategies to inactivate TcdA for the treatment of CDI.

# Materials and Methods

### Protein expression and purification

TcdA produced by the VPI10463 strain of *C. difficile* was used throughout this study. The genes encoding DRBD-5CROPs (residues 843–2,481), DRBD-5CROPs$^{G1241D}$, and DRBD-5CROPs$^{dCys7281}$ were cloned into a modified pET28a vector with a 6 × His/SUMO tag introduced to the N-terminus. The genes encoding DRBD-5CROPs$^{SC}$, DRBD-5CROPs$^{EC}$, the full length TcdA (TcdA$^{WT}$), TcdA$^{G1241D}$, and TcdA$^{dCys7281}$ were cloned into a modified pET22b vector, which have an N-terminal Twin-Strep tag followed by a PreScission protease-cleavage site as well as a C-terminal 6 × His tag. All the mutants were generated by QuikChange PCR and verified by DNA sequencing.

All these proteins were expressed in *Escherichia coli* strain BL21-star (DE3) (Invitrogen). Bacteria were cultured at 37°C until OD$_{600}$ reached 0.6~0.8, and protein expression was then induced with 1 mM isopropyl $\beta$-D-1-thiogalactopyranoside (IPTG). For the expression of DRBD-5CROPs, DRBD-5CROPs$^{G1241D}$, DRBD-5CROPs$^{dCys7281}$, DRBD-5CROPs$^{SC}$, and DRBD-5CROPs$^{EC}$, the temperature was reduced to 18°C for an overnight induction. For the expression of TcdA$^{WT}$, TcdA$^{G1241D}$, and TcdA$^{dCys7281}$, the bacterial culture was kept at 37°C post induction for another 5 h. The bacteria were harvested by centrifugation at 6,000$g$ and stored at –80°C until future use.

DRBD-5CROPs used for crystallization was purified using Ni-NTA (QIAGEN) affinity resins in a buffer containing 50 mM Tris, 400 mM NaCl, and 30 mM imidazole, pH 8.5, and the bound proteins were eluted with a similar buffer containing 400 mM imidazole, pH 8.5. The His/SUMO tag was cleaved by SUMO-protease, which left an artificially introduced serine at the N-terminus of DRBD-5CROPs. This protein was then exchanged to a buffer containing 20 mM Tris, 40 mM NaCl, pH 8.5, and subjected to MonoQ ion-exchange chromatography (GE Healthcare) using a NaCl gradient. The peak fractions were pooled, concentrated, and further purified by gel filtration using a Superdex-200 column (GE Healthcare) in a buffer containing 20 mM Tris and 150 mM NaCl, pH 8.5. The purified DRBD-5CROPs was concentrated to ~10 mg/ml for crystallization and stored at –80°C. DRBD-5CROPs$^{G1241D}$ and DRBD-5CROPs$^{dCys7281}$ were purified using the same protocol.

The proteins used for smFRET studies (DRBD-5CROPs$^{SC}$ and DRBD-5CROPs$^{EC}$) were purified using Ni-NTA (QIAGEN) affinity resins

using a protocol similar to that of the WT protein except that 1 mM TCEP was included in all buffers. These proteins were further purified using Strep-Tactin (IBA) affinity resins. The bound proteins were eluted by 20 mM HEPES, 150 mM NaCl, 50 mM D-biotin, and 1 mM TCEP, pH 7.5, and further purified by gel filtration in PBS and 5 mM TCEP, pH 7.4.

TcdA$^{WT}$, TcdA$^{G1241D}$, and TcdA$^{dCys7281}$ were first purified using Ni-NTA (QIAGEN) affinity resins as described above (no TCEP). These partially purified proteins were captured by Strep-Tactin resins, and 3C protease were then added to the column to cleave the N-terminal Twin-Strep tag and release target proteins. TcdA variants in the flow through was collected, and the GST-tagged 3C protease was removed using glutathione agarose resins (Genesee Scientific). The final products were exchanged to PBS buffer.

### Crystallization

The initial crystallization conditions were identified using a Gryphon crystallization robot (Art Robbins Instruments) with sparse matrix screens (Hampton Research and QIAGEN) at 18°C. Crystal optimizations were carried out using the hanging-drop vapor diffusion method at 18°C by mixing equal volume of protein and reservoir solutions. For DRBD-5CROPs, the best crystals were obtained by streak-seeding in a condition containing 0.1 M Bis-Tris, pH 6.3, and 0.6 M lithium sulfate (final pH at 6.7). Crystals were cryoprotected in 2.2 M lithium sulfate and flash frozen in liquid nitrogen. The crystals of DRBD-5CROPs$^{dCys7281}$ were obtained by streak-seeding in a condition containing 0.1 M Bis-Tris propane (pH 7.0) and 1.0 M Ammonium citrate tribasic (pH 7.0), which were cryoprotected and flash frozen in 1.5 M lithium sulfate.

### Data collection and structure determination

The X-ray diffraction data were collected at 100 K at the NE-CAT beamline 24-ID-C, Advanced Photon Source. The data were processed with XDS software as implemented in RAPD (https://github.com/RAPD/RAPD) (Kabsch, 2010).

To determine the structure of DRBD-5CROPs, we first determined the structure of a fragment containing residues 1,040–1,802 using molecular replacement (PHENIX.Phaser-MR) (McCoy et al, 2007) with the corresponding region in the structure of TcdA$^{1832}$ (PDB code: 4R04) as a search model (Chumbler et al, 2016). Using this partial structure as a fixed partial model, we defined the structure of the CROPs I and II using the CROPs I and II of TcdB (residues 1,834–2,100, PDB code: 4NP4, corresponding to residues 1,832–2,099 in TcdA-VPI10463) as a search model in molecular replacement (Orth et al, 2014). After that, the positions of CROPs III and V were located by another round of molecular replacement using the CROPs VI of TcdA from strain 48,489 (residues 34–146, PDB code: 2G7C, corresponding to residues 2,482–2,594 in TcdA-VPI10463) as a search model (Greco et al, 2006). This partial structure was then used as a fixed partial model to locate the CROPs IV by another round of molecular replacement using the CROPs VI of TcdA-48489 as a search model.

After the structure of residues 1,040–1,802 and the whole CROPs were defined, we used them as a fixed partial model in another round of molecular replacement to locate the SGS using the SGS of

TcdA[1832] as a search model (Chumbler et al, 2016). The structure of DRBD-5CROPs[dCys7281] was solved by molecular replacement using WT DRBD-5CROPs as the search model. Structural modeling and refinement were carried out iteratively using PHENIX.refine (Adams et al, 2010) and COOT (Emsley & Cowtan, 2004). All the refinement progress was monitored with the free R value using a 5% randomly selected test set (Brunger, 1992). The structure was validated by MolProbity (Chen et al, 2010). Table S1 shows the detailed statistics of data collection and refinement of DRBD-5CROPs. All structure figures were prepared using PyMOL (DeLano Scientific) and UCSF Chimera (Pettersen et al, 2004). Calculation of the buried molecular surface area was carried out using PISA (Proteins, Interfaces, Structures and Assemblies) program (Krissinel & Henrick, 2007).

### Trypsin digestion assay

The limited trypsin digestion assays were performed on DRBD-5CROPs, DRBD-5CROPs[G1241D], and DRBD-5CROPs[dCys7281] at two different conditions: (1) 20 mM HEPES, pH 7.4, and 250 mM NaCl; and (2) 20 mM sodium citrate, pH 5.4, and 250 mM NaCl. These proteins at 0.5 mg/ml concentration were mixed with trypsin at a molar ratio of 1:50, and the reaction mixtures were incubated at room temperature (21°C). Samples taken at the indicated time were boiled for 5 min in SDS–PAGE loading buffer to quench the reaction, which were then examined by SDS–PAGE and visualized using Coomassie blue staining.

### In-gel digestion and LC MS/MS analysis

Bands on the SDS–PAGE gel corresponding to fragments F1, F2 and F3 were excised, reduced with TCEP at final concentration of 20 mM, alkylated using iodoacetamide at a final concentration of 40 mM, and digested in-gel with trypsin at 37°C overnight. After extraction, the digested peptides were analyzed by LC MS/MS on an LTQ-Orbitrap XL mass spectrometer (Thermo Fisher Scientific) coupled on-line with an EASY-nLC-1000 (Thermo Fisher Scientific). Peptides were separated on a 20 cm × 100 $\mu$m column packed with Reprosil-Pur C18-AQ, 1.9 $\mu$m resin (Dr. Maisch GmbH). For MS/MS analysis of peptides, a cycle of one full FT scan (350–1,800 m/z, resolution of 30,000 at m/z 400) was followed by 10 data-dependent MS/MS acquired in the LTQ with normalized collision energy set at 29%.

### Peptide mapping

MS/MS data were extracted and subjected to a database search against 1,640 amino acid residues peptide (including an artificially introduced serine at the N-terminus) using a developmental version of Protein Prospector (v5.10.10, University of California San Francisco). The mass tolerances for parent ions and fragment ions were set as ± 20 ppm and 0.6 D, respectively. Trypsin was set as the enzyme with two maximum missed cleavages allowed. Protein N-terminal acetylation, N-terminal conversion of glutamine to pyroglutamic acid and methionine oxidation were selected as variable modifications; cysteine carbamidomethylation was specified as a fixed modification. The minimum score and maximum expectation value were set as 22 and 0.05 for peptide identification. All identified peptides were then quantified at the MS level by

measuring the area of their corresponding precursor and first two isotope peaks using Skyline (v21.10.1.0.146, University of Washington). The exported peptide abundances were then used to determine the abundances of individual residues, which were then normalized within each sample to determine potential truncation sites.

### Cytopathic cell-rounding assay

The cytopathic effects of TcdA[WT], TcdA[G141D], and TcdA[dCys7281] were analyzed using the standard cell-rounding assay. In brief, cells were seeded into 96-well plates and exposed to toxins for 24 h. The phase-contrast images of cells were recorded (Olympus IX51, ×10 to ×20 objectives). A zone of 300 × 300 $\mu$m was selected randomly (containing 50–150 cells). The numbers of normal and round-shaped cells were counted manually, and the data were analyzed using the Excel and Origin software.

### Single-molecule FRET analysis of DRBD-5CROPs

Proteins were randomly labeled with an equimolar ratio of Alexa Fluor 555 C5 maleimide and Alexa Fluor 647 C2 maleimide (Invitrogen) overnight at 4°C. Unconjugated dye was removed by desalting with a PD10 column (Cytiva) followed by dialysis. The labeling efficiency was determined using UV-VIS spectroscopy to be greater than 90%. Fluorescently labeled variants of TcdA were biotinylated by EZ-Link NHS-PEG4-Biotin (Thermo Fisher Scientific) in fivefold molar excess at 4°C overnight in 50 mM potassium phosphate buffer at pH 6.5 to bias labeling towards the N-terminus. Unreacted biotin was removed by desalting with a PD10 column.

Quartz slides were passivated with biotinylated-BSA along with a mixture of Biolipidure 203 and 206 (NOF AMERICA Corporation). Biotinylated proteins were attached via streptavidin at ~100 pM in 50 mM HEPES, 100 mM NaCl, pH 7, to achieve optically resolved single molecules. Under these conditions, there was negligible non-specific sticking to the surface. Images were acquired using a prism-based Total Internal Reflection Fluorescence microscope constructed on an IX71 Olympus base with a 60×/1.2-NA water-immersion objective. Fluorescence emission was spectrally separated using an Optosplit Image splitter (Cairn Research) and collected at 10 Hz using an Andor iXon EMCCD camera (Andor Technologies). Data were collected in either 50 mM HEPES, 100 mM NaCl, pH 7, or 50 mM acetate and 100 mM NaCl, pH 5. Imaging buffers contained 0.8% glucose, 25 U/ml pyranose oxidase, 250 U/ml catalase, and 0.1 mM cyclooctatetraene to forestall photobleaching and prevent blinking. Alternating illumination using 637 nm (Coherent Inc.) and 532 nm (Laser Quantum) lasers allowed for the identification of molecules containing one donor and one acceptor.

Microscopy images were analyzed using in-house MATLAB scripts to correlate donor and acceptor images, extract single molecule intensity time traces and calculate FRET efficiency (McCann et al, 2010). Statistical representations of time-averaged FRET distribution of the states of these variants were fitted with multipeak Gaussian distribution function. For extracting the dwell time values, we used the ebFRET software package (van de Meent et al, 2014).

## Data Availability

The coordinates and structure factors for DRBD-5CROPs has been deposited to the Protein Data Bank under access code 7U1Z. All other relevant data are within the artilce and the Expanded View.

## Supplementary Information

## Acknowledgements

This work was partly supported by National Institute of Allergy and Infectious Diseases (NIAID) grants R01AI125704, R01AI158503, R21AI156092, and R21AI163178 to R.J., R01AI132387 to M.D., R01AI139087 to R.J. and M. D., and National Institute of Mental Health (NIMH) grant R01MH081923 to M.E.B., National Institute of General Medical Sciences (NIGMS) grants R01GM074830 and R01GM130144 to L.H. M.D. holds the Investigator in the Pathogenesis of Infectious Disease award from the Burroughs Wellcome Fund. NE-CAT at the Advanced Photon Source (APS) is supported by a grant from the National Institute of General Medical Sciences (P30 GM124165). Use of the APS, an Office of Science User Facility operated for the U.S. Department of Energy (DOE) Office of Science by Argonne National Laboratory, was supported by the U.S. DOE under Contract No. DE-AC02-06CH11357.

### Author Contributions

B Chen: conceptualization, data curation, formal analysis, validation, investigation, methodology, and writing—original draft, review, and editing.
S Basak: data curation, formal analysis, investigation, methodology, and writing—review and editing.
P Chen: data curation, formal analysis, investigation, methodology, and writing—review and editing.
C Zhang: data curation, formal analysis, investigation, methodology, and writing—review and editing.
K Perry: data curation, formal analysis, investigation, methodology, and writing—review and editing.
S Tian: data curation, formal analysis, investigation, methodology, and writing—review and editing.
C Yu: data curation, formal analysis, investigation, methodology, and writing—review and editing.
M Dong: data curation, formal analysis, supervision, funding acquisition, investigation, methodology, and writing—review and editing.
L Huang: data curation, formal analysis, funding acquisition, investigation, methodology, and writing—review and editing.
ME Bowen: data curation, formal analysis, supervision, funding acquisition, validation, investigation, visualization, methodology, project administration, and writing—review and editing.
R Jin: conceptualization, resources, data curation, formal analysis, supervision, funding acquisition, validation, investigation, visualization, methodology, project administration, and writing—original draft, review, and editing.

### Conflict of Interest Statement

The authors declare that they have no conflict of interest.

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
