## [Reviewer comments · Life Science Alliance]

Life Science Alliance

Structure and conformational dynamics of *Clostridioides difficile* toxin A

Baohua Chen, Sujit Basak, Peng Chen, Changcheng Zhang, Kay Perry, Songhai Tian, Clinton Yu, Min Dong, Lan Huang, Mark Bowen, and Rongsheng Jin

DOI: <https://doi.org/10.26508/lsa.202201383>

Corresponding author(s): Rongsheng Jin, University of California, Irvine and Mark Bowen, Stony Brook University

Review Timeline:

Submission Date:	2022-01-21
Editorial Decision:	2022-02-17
Revision Received:	2022-02-23
Editorial Decision:	2022-02-25
Revision Received:	2022-02-25
Accepted:	2022-02-28

Scientific Editor: Novella Guidi

Transaction Report:

February 17, 2022

Re: Life Science Alliance manuscript #LSA-2022-01383-T

Rongsheng Jin
University of California, Irvine

Dear Dr. Jin,

Thank you for submitting your manuscript entitled "Structure and conformational dynamics of Clostridioides difficile toxin A" to Life Science Alliance. The manuscript was assessed by expert reviewers, whose comments are appended to this letter. We, thus, encourage you to submit a revised version of the manuscript back to LSA that responds to all of the reviewers' points.

Thank you for this interesting contribution to Life Science Alliance. We are looking forward to receiving your revised manuscript.

Sincerely,

B. MANUSCRIPT ORGANIZATION AND FORMATTING:

Reviewer #1 (Comments to the Authors (Required)):

The manuscript

Structure and conformational dynamics of Clostridioides difficile toxin A

by Chen and colleagues describes crystallographic, biochemical, biophysical and cell-biology work with TcdA, one of the large toxins of the pathogenic bacterium *C. difficile*. With over 2700 amino acids, TcdA is the "little" brother of TcdB, and high-resolution structure determination of more or less complete TcdA/B has only recently been achieved in a handful of landmark papers. This breakthrough was also fueled by the better availability of cryoEM methods.

Here, the authors use protein crystallography to determine a neutral pH structure of a TcdA fragment that lacks two enzyme domains at the N-terminus (residues 843 to 2481) but is slightly more extended into the C-terminal repetitive "CROP"-domains-containing region than previously published structures. The structure provides detailed insight into the lasso-loop-like interaction between the CROPs-tail and the delivery and receptor-binding domain (DRBD) of TcdA. This seems to keep the toxin in a closed state required to e.g. protect it from protease degradation, and it seems to open up at lower pH when the toxin resides in endosomes and initiates membrane translocation of the glycosylating domain that is at the heart of TcdA's and TcdB's toxicity. Such a higher resolution closed-form structure of TcdA or TcdB was not available until very recently, and the work presented here provides important insight into the structural principles of keeping these toxins in the closed state.

Being a crystallographer myself, I understand that the crystal structure determination reported here required very high skills and was extremely laborious to achieve. I am therefore certain that a recent publication of a similar cryoEM structure of TcdA by a group from Denmark (Aminzadeh et al., 2022; by coincidence I was a referee for this paper as well) must have come as a shock to Chen and colleagues. The present manuscript, however, takes our insight into the structural dynamics of TcdA (and likely TcdB as well) further by employing additional methods such as protease sensitivity and single-molecule FRET experiments to corroborate pH-induced movements within TcdA. Finally, the authors show that toxicity of TcdA is strongly reduced when the toxin is blocked in the closed state.

Overall, the work by Chen et al. is an important contribution for the field that I enjoyed reading, and I have only very few remarks that I suggest changing to improve the manuscript:

- In general, it would have been helpful if the manuscript would have come with line and page numbers
- The reference to the recently published paper by Aminzadeh needs to be updated
- In their discussion of the structure, the authors should more openly indicate if their analysis comes to the same conclusions as the Aminzadeh paper and where it deviates
- In Fig. 6A, the small cyan dots are easily overlooked - please consider changing it to something more visible
- The final sentence in the paragraph "The interplay between the DRBD and the CROPs of TcdA" reads a little strange, and the reference to Krissinel & Henrick, 2007 sounds as if these authors have done an analysis of TcdA - consider rewording
- I could not really make out if "CROPs" is a plural or singular, I would guess it is plural. Consequently, the "undergoes" in the chapter "pH-dependent dynamics of TcdA" should be changed to "undergo" - I may be wrong here.
- Data availability: the authors indicate that they have not yet deposited the structure of their TcdA fragment. I think that this is not acceptable, and that a PDB entry code must be provided.

Reviewer #2 (Comments to the Authors (Required)):

The authors report on the crystal structure of a fragment of *C. difficile* toxin A (TcdA) at 3.18 Å resolution, covering residues 843 through 2481, including the receptor binding and translocation domain DRBD and the CROPs domain parts I-V. In addition, they present data on limited proteolysis studies, which were performed at neutral (pH 7.4) and acidic pH, and data of single molecule FRET analyses, which support dynamic changes of the toxin structure at low pH. This is an interesting paper, which is in line with recent publications. Especially, the recent publication on the cryo-EM structure of TcdA (Aminzadeh et al. EMBO reports 2022) has addressed many of the questions studied here. Therefore, novelty and impact of the data are limited.

Specific comments:

1. In general, the recent manuscript by Aminzadeh et al. should be discussed in comparison to data presented here more extensively.
2. Sometimes, it is not clear what is new and what is from former studies. Moreover, if data are from modeling this should clearly be stated. For example, dynamic changes of toxin structures are not directly observed in this study, therefore a statement (page 9) like "We found that the SGS domain in the DRBD...." is not really supported by experimental data.
3. Abstract: "Here, we report ..., which fills the knowledge gap...". Actually I could not really see which "knowledge gap" e.g. in comparison to the paper by Aminzadeh et al. is filled. Please change the wording.
4. The citation of Aminzadeh et al. should include the publication year.
5. The first paragraph of the results part can be omitted, because the information is not important. It might be partly included into the Introduction.
6. Please, give the *C. difficile* strain from which the toxin comes also under Material and Methods.
7. What is known about the impact of the flexible region covering residues 1661-1667.
8. Statement (page 4): ".....resulting the first complete structure of TcdA holotoxin...." This is a modeling and has been done by others, too.
9. The presentation and discussion of the structural data is very similar to those of the paper by Aminzadeh et al.. Therefore, these data should be compared with the present data.
10. Page 6: The interactions at interface II are partly different as compared to the paper by Aminzadeh et al.. this should be stated and discussed.
11. Discussion (page 10): The paragraph starting with "While the" needs additional references and, again, a discussion about the results obtained by Aminzadeh et al.

Reviewer #3 (Comments to the Authors (Required)):

This manuscript by Chen, et al. presents structural and biochemical data probing the conformational dynamics of TcdA at both neutral and acidic pH. The X-ray crystal structure of a TcdA fragment was determined, covering the majority of the delivery and receptor binding domain (DRBD), including an N-terminal region the authors label the small globular subdomain (SGS), along with CROPs repeats I-V. With this new structural data, the authors assemble a complete model of the TcdA holotoxin. Importantly, the authors contrast their structure, determined at neutral pH, with existing TcdA structures (Aminzadeh, et al. 2021; Pruitt, et al. 2010), and with the TcdB holotoxin structure, determined at low pH. Although previously established that TcdA and other large clostridial family toxins undergo drastic pH-dependent conformational rearrangements, the authors identify specific structural states (open and closed) that the toxin may sample under endosomal pH. These were confirmed by limited trypsin digestion at pH 7.4 and pH 5.4, identification of trypsin cleavage products by mass spectrometry, structure-based site-directed mutagenesis to generate TcdA mutants that either disrupt and strengthen interactions between the DRBD and CROPs, single-molecule FRET to establish TcdA's pH-dependent dynamics, and lastly, cell-rounding assays to test the cytotoxicity of TcdA mutants that influence the flexibility of the CROPs. The authors conclude that the SGS of the DRBD undergoes coordinated conformational changes with the CROPs to protect the pore-forming region from proteolysis at neutral pH, and allow for exposure of the pore-forming region at endosomal pH.

The manuscript is well-written and easy to follow. The experiments performed were thorough and considerate of outstanding questions concerning TcdA dynamics and how the information generated by these studies may inform future efforts to neutralize TcdA through limiting its structural plasticity. All claims are well supported by the data.

Major Concerns:

None.

Minor Issues - Scientific:

Table indicate that there were 180 ligand/ions and 0 waters. Presumably a mix-up?

Minor Issues - Editorial:

Abstract: line 4, change 'understating' to 'understanding'

- Page 3, line 16: change to something like "and that conformation dynamics is crucial for TcdA cytotoxicity."
- Page 5, line 13: change compared to comparing
- p.7: change 'subsequence' actions to 'subsequent' actions
- Page 8, line 13: delete the 2nd "that"
- Page 9, line 5: delete "once occurs"
- Page 9, 1st and 2nd lines of last paragraph: delete "domain" (small globular subdomain domain is redundant)
- Page 11, line 4: change to "our structural modeling suggests that antibodies binding.."
- Figure EV1 A caption: green has an extra "e"
- Figure EV1 B caption: separately is misspelled

Life Science Alliance manuscript #LSA-2022-01383-T**Structure and conformational dynamics of *Clostridioides difficile* toxin A**

We thank the editors and reviewers for their careful reading of the manuscript and their constructive suggestions that have guided our revision. We have revised the manuscript to address reviewers' concerns as outlined below. We would like to point out that our studies presented here were carried out independently, using different experiments and analyses, from the beautiful cryoEM studies recently published by Jørgensen and colleagues (Aminzadeh *et al.* EMBO reports 2022). Our crystal structure was determined back in 2019 but we decided to continue with more mechanistic studies, which is our ultimate goal when resolving a protein structure. Unfortunately, these studies were delayed by COVID restrictions. As Reviewer #1 pointed out, the recent structure paper did come as a shock to us when we were preparing our manuscript. We are glad that our two groups arrived at similar conclusions regarding the structure of TcdA and in fact corroborate each other's findings. We cite Aminzadeh *et al.* throughout our paper, and have added more structural comparisons in the revised manuscript. However, we hope the reviewers appreciated that we have presented our data as an independent work, with a neutral point of view, instead of focusing on comparison with the recent cryoEM structure.

Reviewer #1

The manuscript, Structure and conformational dynamics of Clostridioides difficile toxin A, by Chen and colleagues describes crystallographic, biochemical, biophysical and cell-biology work with TcdA, one of the large toxins of the pathogenic bacterium C. difficile. With over 2700 amino acids, TcdA is the "little" brother of TcdB, and high-resolution structure determination of more or less complete TcdA/B has only recently been achieved in a handful of landmark papers. This breakthrough was also fueled by the better availability of cryoEM methods.

Here, the authors use protein crystallography to determine a neutral pH structure of a TcdA fragment that lacks two enzyme domains at the N-terminus (residues 843 to 2481) but is slightly more extended into the C-terminal repetitive "CROP"-domains-containing region than previously published structures. The structure provides detailed insight into the lasso-loop-like interaction between the CROPs-tail and the delivery and receptor-binding domain (DRBD) of TcdA. This seems to keep the toxin in a closed state required to e.g. protect it from protease degradation, and it seems to open up at lower pH when the toxin resides in endosomes and initiates membrane translocation of the glycosylating domain that is at the heart of TcdA's and TcdB's toxicity. Such a higher resolution closed-form structure of TcdA or TcdB was not available until very recently, and the work presented here provides important insight into the structural principles of keeping these toxins in the closed state.

*Being a crystallographer myself, I understand that the crystal structure determination reported here required very high skills and was extremely laborious to achieve. I am therefore certain that a recent publication of a similar cryoEM structure of TcdA by a group from Denmark (Aminzadeh *et al.*, 2022; by coincidence I was a referee for this paper as well) must have come as a shock to Chen and colleagues. The present manuscript, however, takes our insight into the structural dynamics of TcdA (and likely TcdB as well) further by employing additional methods such as protease sensitivity and single-molecule FRET experiments to corroborate pH-induced movements within TcdA. Finally, the authors show that toxicity of TcdA is strongly reduced when the toxin is blocked in the closed state.*

*Overall, the work by Chen *et al.* is an important contribution for the field that I enjoyed reading, and I*

have only very few remarks that I suggest changing to improve the manuscript:

- In general, it would have been helpful if the manuscript would have come with line and page numbers

Response: We have added the line and page numbers as suggested.

- The reference to the recently published paper by Aminzadeh needs to be updated

Response: This reference has been updated.

- In their discussion of the structure, the authors should more openly indicate if their analysis comes to the same conclusions as the Aminzadeh paper and where it deviates

Response: We have included more discussion and comparisons with the structure reported by Aminzadeh et al.

- In Fig. 6A, the small cyan dots are easily overlooked - please consider changing it to something more visible

Response: We have changed the cyan dots to magenta.

- The final sentence in the paragraph "The interplay between the DRBD and the CROPs of TcdA" reads a little strange, and the reference to Krissinel & Henrick, 2007 sounds as if these authors have done an analysis of TcdA - consider rewording

Response: Sorry for the confusion. This reference was cited here for the PISA program we used. We have deleted this reference from this sentence and moved it to the Method: "Calculation of the buried molecular surface area was carried out using PISA (Proteins, Interfaces, Structures and Assemblies) program (Krissinel & Henrick, 2007)." (Lines 512-514, Page 12)

- I could not really make out if "CROPs" is a plural or singular, I would guess it is plural. Consequently, the "undergoes" in the chapter "pH-dependent dynamics of TcdA" should be changed to "undergo" - I may be wrong here.

Response: It has been changed to "undergo".

- Data availability: the authors indicate that they have not yet deposited the structure of their TcdA fragment. I think that this is not acceptable, and that a PDB entry code must be provided.

Response: The structure has been deposited to PDB under access code 7U1Z.

Reviewer #2

The authors report on the crystal structure of a fragment of C. difficile toxin A (TcdA) at 3.18 Å resolution, covering residues 843 through 2481, including the receptor binding and translocation domain DRBD and the CROPs domain parts I-V. In addition, they present data on limited proteolysis studies, which were performed at neutral (pH 7.4) and acidic pH, and data of single molecule FRET analyses, which support dynamic changes of the toxin structure at low pH. This is an interesting paper, which is in line with recent publications. Especially, the recent publication on the cryo-EM structure of TcdA

(Aminzadeh et al. EMBO reports 2022) has addressed many of the questions studied here. Therefore, novelty and impact of the data are limited.

Specific comments:

1. In general, the recent manuscript by Aminzadeh et al. should be discussed in comparison to data presented here more extensively.

Response: As suggested, we have included more discussion and comparisons with the structure reported by Aminzadeh et al. The individual changes have been tracked in the revised manuscript.

2. Sometimes, it is not clear what is new and what is from former studies. Moreover, if data are from modeling this should clearly be stated. For example, dynamic changes of toxin structures are not directly observed in this study, therefore a statement (page 9) like "We found that the SGS domain in the DRBD...." is not really supported by experimental data.

Response: We rephrased this sentence, which now reads as "Structural analysis showed that the SGS in the DRBD ...". The 3rd sentence in this paragraph reads as "Our structure model suggests that ...".

3. Abstract: "Here, we report ..., which fills the knowledge gap...". Actually I could not really see which "knowledge gap" e.g. in comparison to the paper by Aminzadeh et al. is filled. Please change the wording.

Response: We changed it to "Here, we report ..., which advances our understanding of ...".

4. The citation of Aminzadeh et al. should include the publication year.

Response: Updated.

5. The first paragraph of the results part can be omitted, because the information is not important. It might be partly included into the Introduction.

Response: The purpose of this paragraph is to explain the rationale for focusing on a fragment of TcdA covering the DRBD and CROPs. We feel this information is better presented at this point in the manuscript, which naturally leads to experimental details of protein production and structure determination.

6. Please, give the *C. difficile* strain from which the toxin comes also under Material and Methods.

Response: We added this information to the Method: "TcdA produced by the VPI10463 strain of *C. difficile* was used throughout this study."

7. What is known about the impact of the flexible region covering residues 1661-1667.

Response: This region is part of a loop that interacts with the GTD, which is disordered in our structure in the absence of the GTD.

8. Statement (page 4): ".....resulting the first complete structure of TcdA holotoxin...." This is a modeling and has been done by others, too.

Response: We have rephrased it to "...resulting a complete structural model of TcdA holotoxin...".

9. *The presentation and discussion of the structural data is very similar to those of the paper by Aminzadeh et al.. Therefore, these data should be compared with the present data.*

Response: As noted above, we have included more discussion and comparisons with the structure reported by *Aminzadeh et al.* The individual changes have been tracked in the revised manuscript.

We are glad that these two reports arrived at many similar conclusions regarding the structure of TcdA that corroborate each other's findings. Because these two projects were developed independently, we feel it is appropriate to present our data in an independent and neutral manner, as opposed to focusing on point-by-point comparisons with the cryoEM structure. This way, the audiences will read both stories and draw independent conclusions.

10. *Page 6: The interactions at interface II are partly different as compared to the paper by Aminzadeh et al.. this should be stated and discussed.*

Response: The interface II is largely identical in two structures, except for some subtle differences in side chain orientation. For example, the side chain of R2260, which has good electron density in our structure, forms a hydrogen bond with T1278. However, this same side chain adopts a different rotamer position in the cryoEM structure and instead forms salt bridges with E1112 and E1235. These differences are likely due to the different conditions and methods used for structure determination.

11. *Discussion (page 10): The paragraph starting with "While the" needs additional references and, again, a discussion about the results obtained by Aminzadeh et al.*

Response: We added additional discussion including the results obtained by *Aminzadeh et al.* (lines 397-404, page 10).

Reviewer #3

*This manuscript by Chen, et al. presents structural and biochemical data probing the conformational dynamics of TcdA at both neutral and acidic pH. The X-ray crystal structure of a TcdA fragment was determined, covering the majority of the delivery and receptor binding domain (DRBD), including an N-terminal region the authors label the small globular subdomain (SGS), along with CROPs repeats I-V. With this new structural data, the authors assemble a complete model of the TcdA holotoxin. Importantly, the authors contrast their structure, determined at neutral pH, with existing TcdA structures (*Aminzadeh, et al. 2021; Pruitt, et al. 2010*), and with the TcdB holotoxin structure, determined at low pH. Although previously established that TcdA and other large clostridial family toxins undergo drastic pH-dependent conformational rearrangements, the authors identify specific structural states (open and closed) that the toxin may sample under endosomal pH. These were confirmed by limited trypsin digestion at pH 7.4 and pH 5.4, identification of trypsin cleavage products by mass spectrometry, structure-based site-directed mutagenesis to generate TcdA mutants that either disrupt and strengthen interactions between the DRBD and CROPs, single-molecule FRET to establish TcdA's pH-dependent dynamics, and lastly, cell-rounding assays to test the cytotoxicity of TcdA mutants that influence the flexibility of the CROPs. The authors conclude that the SGS of the DRBD undergoes coordinated conformation changes with the CROPs to protect the pore-forming region from proteolysis at neutral pH, and allow for exposure of the pore-forming region at endosomal pH.*

The manuscript is well-written and easy to follow. The experiments performed were thorough and considerate of outstanding questions concerning TcdA dynamics and how the information generated by

these studies may inform future efforts to neutralize TcdA through limiting its structural plasticity. All claims are well supported by the data.

Major Concerns:

None.

Minor Issues - Scientific:

Table indicate that there were 180 ligand/ions and 0 waters. Presumably a mix-up?

Response: At the current resolution, we observed 36 molecules of sulfate in our structure, but no well-defined waters. This is likely due to the presence of 0.6 M lithium sulfate in the crystallization mother liquor and the crystals were cryoprotected in 2.2 M lithium sulfate.

Minor Issues - Editorial:

Abstract: line 4, change 'understating' to 'understanding' – (line 20, page 1)

- Page 3, line 16: change to something like "and that conformation dynamics is crucial for TcdA cytotoxicity." – (line 87, page 3).
- Page 5, line 13: change compared to comparing – (line 169, page 5).
- p.7: change 'subsequence' actions to 'subsequent' actions – (line 274, page 7).
- Page 8, line 13: delete the 2nd "that" – (line 303, page 8).
- Page 9, line 5: delete "once occurs" – (line 339, page 9).
- Page 9, 1st and 2nd lines of last paragraph: delete "domain" (small globular subdomain domain is redundant) – (lines 371, 373, page 10).
- Page 11, line 4: change to "our structural modeling suggests that antibodies binding.." – (line 428, page 11).
- Figure EV1 A caption: green has an extra "e" – Changed.
- Figure EV1 B caption: separately is misspelled – Changed.

Response: Thank you so much for your careful reading, and we have made changes as suggested. The line and page numbers in the track changed manuscript are listed in parentheses.

February 25, 2022

RE: Life Science Alliance Manuscript #LSA-2022-01383-TR

Dr. Rongsheng Jin
University of California, Irvine
821 Health Sciences Road
Medical Sciences C, C333
Irvine, CA 92697

Dear Dr. Jin,

Thank you for submitting your revised manuscript entitled "Structure and conformational dynamics of Clostridioides difficile toxin A". We would be happy to publish your paper in Life Science Alliance pending final revisions necessary to meet our formatting guidelines.

- Please upload all figure files as individual ones, including the supplementary figure files; all figure legends should only appear in the main manuscript file
- please upload your Tables in editable .doc or excel format;
- please add the supplementary references to the main references list
- please add ORCID ID for secondary corresponding-they should have received instructions on how to do so
- please add a callout for Figure S2D to your main manuscript text

A. FINAL FILES:

B. MANUSCRIPT ORGANIZATION AND FORMATTING:

Sincerely,

February 28, 2022

RE: Life Science Alliance Manuscript #LSA-2022-01383-TRR

Dr. Rongsheng Jin
University of California, Irvine
821 Health Sciences Road
Medical Sciences C, C333
Irvine, CA 92697

Dear Dr. Jin,

Thank you for submitting your Research Article entitled "Structure and conformational dynamics of Clostridioides difficile toxin A". It is a pleasure to let you know that your manuscript is now accepted for publication in Life Science Alliance. Congratulations on this interesting work.

DISTRIBUTION OF MATERIALS:

Again, congratulations on a very nice paper. I hope you found the review process to be constructive and are pleased with how the manuscript was handled editorially. We look forward to future exciting submissions from your lab.

Sincerely,
